# High Triglyceride-Glucose Index with Renal Hyperfiltration and Albuminuria in Young Adults: The Korea National Health and Nutrition Examination Survey (KNHANES V, VI, and VIII)

**DOI:** 10.3390/jcm11216419

**Published:** 2022-10-29

**Authors:** Donghwan Oh, Sang Ho Park, Seoyoung Lee, Eunji Yang, Hoon Young Choi, Hyeong Cheon Park, Jong Hyun Jhee

**Affiliations:** 1Division of Nephrology, Department of Internal Medicine, Gangnam Severance Hospital, Seoul 06273, Korea; 2Department of Internal Medicine, Yonsei University College of Medicine, Seoul 03722, Korea; 3Severance Institute for Vascular and Metabolic Research, Yonsei University College of Medicine, Seoul 03722, Korea

**Keywords:** TyG index, renal hyperfiltration, insulin resistance, albuminuria, young adult

## Abstract

**Background:** High triglyceride-glucose (TyG) index, a surrogate marker of insulin resistance, is associated with an increased risk of albuminuria in adults. However, the relationship between high TyG index associated with renal hyperfiltration (RHF) and albuminuria among young adults is unclear. **Methods:** A total of 5420 participants aged 19–39 years were enrolled from the Korean National Health and Nutrition Examination Survey (2011–2014 and 2019) and their TyG index levels were analyzed. RHF was defined as eGFR with residuals > 90th percentile after adjusting for age, sex, weight, and height. Albuminuria was defined as urinary albumin-to-creatinine ratio ≥ 30 mg/g Cr. Logistic regression analyses were used to evaluate the association between TyG index, RHF, and albuminuria. **Results:** The mean age was 30.7 ± 6.0 years and 46.4% were male. The prevalence of albuminuria and RHF was higher in the higher tertiles of TyG index. In our multivariable model, high TyG index showed higher risk of albuminuria (odds ratio (OR) per 1.0 increase in TyG index, 1.56; 95% confidence interval (CI), 1.24–1.95 and OR in the highest tertile, 1.65; 95% CI, 1.08–2.52). High TyG index was associated with higher risk of RHF (OR per 1.0 increase in TyG index, 1.56; 95% CI, 1.32–1.84 and OR in the highest tertile, 1.73; 95% CI, 1.31–2.30). When participants were divided into with or without RHF, high-TyG index-associated high risk of albuminuria was only observed in those with RHF. Participants with concurrent high TyG index and RHF showed the highest risk of albuminuria. Mediation analysis showed that 54.2% of the relation between TyG index and albuminuria was mediated by RHF (95% CI of indirect effect, 0.27–0.76). Finally, incorporating TyG index into our basic model improved the predictive value for albuminuria only in participants with RHF. **Conclusion:** High TyG index associated with RHF was the strongest risk factor for albuminuria in this study. Early identification of high TyG index with RHF may prevent future development of CKD in relatively healthy and young adults.

## 1. Introduction

Chronic kidney disease (CKD) is associated with high rates of morbidity and mortality, resulting in a substantial socioeconomic burden [1]. Young adults with CKD are also at increased risk of adverse cardiovascular (CV) complications and mortality, similar to older CKD adults [2,3]. Despite advances in care for CKD, young adults with CKD still suffer from poor outcomes [4]. Given the increasing prevalence of obesity, hypertension, and diabetes in the younger population, the number of young adults with CKD is likely to rise [5,6,7]. However, most young adults are clinically silent and rarely visit the hospital, leaving them in a medical blind spot. Consequently, the incidence of CKD in young adults is frequently underestimated [8]. Furthermore, evidence regarding kidney dysfunction including albuminuria in young adults is relatively scarce, as most studies on CKD use middle-aged and elderly adults [9]. Therefore, increased attention is needed to detect albuminuria early in young adults and to manage modifiable risk factors of CKD.

Insulin resistance is one of the key mechanisms involved in kidney disease development, which is accompanied by renal hyperfiltration (RHF) [10,11,12]. The worsening of insulin resistance increases glomerular capillary pressure and induces glomerular hyperfiltration [13,14]. RHF is an early manifestation of kidney damage and predisposes one to an eventual reduction in glomerular filtration rate (GFR) and overt nephropathy [15,16]. In adolescents and young adults with type 2 diabetes (T2D), a significant association between insulin resistance and RHF was observed in previous clinical studies [17,18]. However, the relationship between insulin resistance and kidney dysfunction mediated through RHF is poorly elucidated in young adults.

The triglyceride-glucose (TyG) index is a surrogate marker of insulin resistance. Recent studies have reported that a high TyG index is associated with insulin resistance and is related to poor kidney outcomes in adults [19,20,21]. However, it is unclear whether a high TyG index is associated with albuminuria in young adults. Therefore, this study aimed to investigate the association between the TyG index with the mediating effect of RHF (an early stage of kidney injury) and albuminuria in young adults.

## 2. Methods

### 2.1. Study Population

Data were taken from the Korean National Health and Nutrition Examination Survey (KNHANES) from 2011–2014 and 2019. The KNHANES is a nationally representative database of the South Korean population controlled by the Korea Centers for Disease Control and Prevention (KCDC, Cheongju, Republic of Korea). The KNHANES database consists of health interviews, health examinations, nutritional surveys, and laboratory tests. All participants were randomly selected from 600 randomly selected districts of cities and provinces in South Korea. In this retrospective cross-sectional study, 43,338 participants from the KNHANES were initially screened. Among them, 37,918 participants were excluded, including those aged ≥40 years or ≤18 years, missing demographic and laboratory data, or with an eGFR below 60 mL/min/1.73 m^2^. Finally, 5420 participants aged 19–39 years with normal kidney function were analyzed (Figure 1). All participants in KNHANES provided written informed consent before the study began. This study adhered to the Declaration of Helsinki and was approved by the Institutional Review Board (IRB) of the KCDC (KNHANES 2010–2014 IRB approval numbers: 2010-02CON-21-C, 2011-02CON-06-C, 2012-01EXP-01-2C, 2013-07CON-03-4C, and 2013-12EXP-03-5C; KNHANES 2019 IRB approval number: 2018-01-03-C-A).

### 2.2. Data Collection

Demographic and socioeconomic data, including age, sex, smoking status, alcohol intake, education, economic status, and medical history, were collected upon study enrollment. Detailed methods for data collection are described in Appendix A. For the laboratory assessment, all participants fasted for at least 8 h before the blood sampling. Serum concentrations of blood urea nitrogen, creatinine, hemoglobin, glucose, total cholesterol, triglyceride, high-density lipoprotein cholesterol (HDL-C), low-density lipoprotein cholesterol (LDL-C), and HbA1c were measured. Serum creatinine was measured using the rate-blanked compensated Jaffe kinetic method with a Hitachi Automatic Analyzer 7600-210 (Hitachi, Tokyo, Japan) from 2010 to 2014 and using a kinetic colorimetric assay (Cobas analyzer; Roche, Mannheim, Germany) in 2019, which both were traceable to the isotope dilution mass spectrometry reference method. The eGFR was calculated using the CKD Epidemiology Collaboration (CKD-EPI) equation [22]. Urine samples were collected in the morning after the first void and spot urine albumin and urine creatinine were measured by Turbidimetric assay (Hitachi Automatic Analyzer 7600, Hitachi, Tokyo, Japan) and the rate-blanked compensated Jaffe kinetic method (Hitachi Automatic Analyzer 7600-210, Hitachi, Tokyo, Japan), respectively. Albuminuria was calculated using the spot urine albumin-to-creatinine ratio (UACR). For detailed methods, see Appendix A.

### 2.3. Definitions of Exposure and Outcome

The TyG index was calculated as ln (fasting triglyceride [mg/dL] × fasting glucose [mg/dL]/2). The study participants were grouped by tertiles based on the TyG index levels (Tertile 1 (lowest), Tertile 2, and Tertile 3 (highest)). RHF was defined using previously suggested methods with some modifications [11]. In brief, the residuals were calculated from a multivariable linear regression analysis in which the logarithm-transformed eGFR was the dependent variable and the logarithm-transformed age, sex, height, and weight were the independent variables. RHF was defined as an absolute logarithm-transformed eGFR > 90th percentile after adjusting for logarithm-transformed age, sex, height, and weight. The primary endpoint was the risk of albuminuria, defined as UACR more than 30 mg/g Cr.

### 2.4. Statistical Analyses

Continuous variables are expressed as the mean ± standard deviation (SD) and categorical variables are expressed as an absolute number (%). Analysis of variance was used to compare continuous variables in each group, and the Chi-squared test was used for categorical variables. The Kolmogorov–Smirnov test was performed to evaluate the normality of the distribution of parameters. If the normal distribution was not verified in the resulting data, the geometric mean ± SD was reported. For multiple comparisons, the Kruskal–Wallis test was used. To further elucidate the association of high TyG index with RHF and the risk of albuminuria, participants were divided according to the presence of RHF and relative risks for albuminuria with TyG index levels and were then compared. Additionally, six groups were made combining the presence or absence of RHF and tertiles of TyG index, and the risks for albuminuria were compared using multivariable logistic analysis. Covariates adjusted for multivariable models included age, sex, smoking and alcohol status, education and income levels, BMI, SBP, histories of hypertension, diabetes, and dyslipidemia, hemoglobin, and eGFR. To investigate the association between TyG index level and albuminuria or RHF, we used multivariable logistic regression analyses. Variance Inflation Factors (VIF) and tolerance for multi-collinearity evaluation with diabetes, BMI, and smoking were performed, and variables with VIF < 10 or tolerance > 0.10 [23,24] were included as covariates (Appendix A). Restricted cubic spline analyses for albuminuria with TyG index levels were performed using the smoothing technique, and the spline degrees of freedom were selected based on the lowest set. To further evaluate the mediation effect of RHF between high TyG index and albuminuria, mediation analysis was performed for the indirect effect of RHF on the TyG index and albuminuria, using the Baron–Kenny mediation method with the Sobel test and bootstrapping with 1000 resamples for confidence intervals. Finally, receiver operating characteristic (ROC) analysis was performed to determine the area under the curve (AUC) of traditional risk factors and the TyG index for predicting albuminuria risk. After fitting the multivariable logistic regression model of each factor and obtaining the model-predicted probability of a positive outcome, two ROC curves were compared using Delong’s test. 

Several sensitivity analyses were performed. First, we performed subgroup analyses to observe interactive effects for RHF and albuminuria between TyG index and diabetes (no vs. yes), smoking (no vs. yes), and BMI (<25 vs. BMI ≥ 25 kg/m^2^). Second, to further evaluate the association of TyG index with RHF and the risk of albuminuria according to glucose levels ((1) prediabetes: 100 ≤ fasting glucose ≤ 125 or 5.7 ≤ HbA1c ≤ 6.4; (2) diabetes: fasting glucose ≥ 126 or HbA1c ≥ 6.5), multivariable logistic regression was performed. Lastly, the definition of RHF using a higher threshold of eGFR with residuals > 95th percentile after adjusting for age, sex, weight, and height was used and analyzed as a study endpoint. For all analyses, a value of *p* < 0.05 was statistically significant. All statistical analyses were conducted using SPSS version 25.0 (IBM Corporation, Chicago, IL, USA), Stata version 17 (StataCorp, College Station, TX, USA), and R software version 4.1.0 (R Project for Statistical Computing, Vienna, Austria).

## 3. Results

### 3.1. Baseline Characteristics

The mean age of the 5420 participants at baseline was 30.7 ± 6.0 years, and 2516 (46.4%) participants were male. The baseline characteristics of study participants according to tertiles of the TyG index are shown in Table 1. The mean levels of the TyG index in each tertile were 7.7 ± 0.3, 8.3 ± 0.2, and 9.1 ± 0.5 in tertiles 1, 2, and 3, respectively. Compared with the lowest tertile, the participants in the higher tertiles were more likely to be older and male and smoke, drink more alcohol, and have lower education or income levels. Participants in the higher tertiles showed higher BMI, SBP, and DBP levels than the lowest tertile. Moreover, the prevalence of hypertension, diabetes, and dyslipidemia was higher in the higher tertiles than the lowest tertile. In laboratory tests, participants in the higher tertiles showed low eGFR levels and higher prevalence of microalbuminuria. The levels of hemoglobin, fasting plasma glucose, HbA1c, and lipid profiles including total cholesterol, triglyceride, and LDL-C were higher. In contrast, the level of HDL-C was lower in the higher tertiles compared to the lowest tertile.

### 3.2. Association of TyG Index with Albuminuria

Of the 5420 participants, albuminuria was prevalent in 233 (4.3%) (Table 2). Tertiles with higher TyG index showed a significantly higher prevalence of albuminuria (44 (2.4%), 64 (3.5%), and 125 (6.9%) cases in tertiles 1, 2, and 3; *p* for trend < 0.001). To investigate the association between TyG index and the risk of albuminuria, logistic regression analysis was performed. High TyG index level was significantly associated with greater odds for albuminuria in the unadjusted model (odds ratio (OR) per 1.0 increase in TyG index level, 2.41; 95% confidence interval (CI), 2.03–2.86; *p* < 0.001). This association remained consistent after full adjustment for confounding variables (adjusted OR (AOR) per 1.0 increase of TyG index level, 1.56; 95% CI, 1.24–1.95; *p* < 0.001). A positive linear relationship between the TyG index levels and the risk of albuminuria was observed by restricted cubic spine curve analysis (Figure 2A). When tertiles of TyG index were evaluated for the risk of albuminuria, the highest tertile was significantly associated with higher risk of albuminuria, with the lowest tertile as the reference group (AOR in Tertile 3, 1.65; 95% CI, 1.08–2.52; *p* = 0.02).

### 3.3. Association of TyG Index with RHF

To elucidate whether high TyG index, which indicates worsening insulin resistance, is associated with kidney injury induced by glomerular hyperfiltration, we tested the association between TyG index and the risk of RHF. Among the study participants, 543 (10%) participants had RHF (Table 3). Based on the TyG index tertile, the higher tertiles had an increased prevalence of RHF (107 (5.9%), 161 (8.9%), and 275 (15.2%) cases in tertiles 1, 2, and 3; *p* for trend < 0.001). To evaluate whether high TyG index is associated with an increased risk of RHF, multivariable logistic regression analysis was conducted. The unadjusted model showed that the risk of RHF increased as the TyG index level increased (OR per 1.0 increase in TyG index level, 2.03; 95% CI 1.79–2.30; *p* < 0.001). In the multivariable model after full adjustment for confounding variables, high TyG index level was consistently associated with an increased risk of RHF (AOR, 1.56; 95% CI 1.32–1.84; *p* < 0.001). Comparing the risk of RHF according to tertiles of TyG index, the highest tertile was significantly associated with an increased risk of RHF compared to the lowest tertile (AOR in Tertile 3, 1.73; 95% CI, 1.31–2.30; *p* < 0.001).

### 3.4. Association between TyG Index with the Presence of RHF and the Risk of Albuminuria

Next, to further identify the association between high TyG index with RHF and the risk of albuminuria, we evaluated the risk of albuminuria according to tertiles of the TyG index with or without RHF (Table 4). When the interaction term was assessed between the risk of albuminuria for TyG index and RHF, a significant interaction was observed (*p* for interaction < 0.001). The prevalence of albuminuria was significantly increased in higher tertile of TyG index in both groups with and without RHF (*p* for trend < 0.001). When the adjusted odds for the albuminuria risk were assessed in participants with RHF, high TyG index was significantly associated with an increased risk of albuminuria (AOR per 1.0 increase in TyG index level, 2.75; 95% CI, 1.74–4.35; *p* < 0.001). However, in participants without RHF, no significant associations were observed between TyG index levels and the risk of albuminuria (AOR per 1.0 increase in TyG index level, 1.23; 95% CI, 0.92–1.65; *p* = 0.16). The restricted cubic spline curve analyses showed a positive linear relationship between TyG index levels and the albuminuria risk only in participants with RHF (Figure 2B,C). When the adjusted odds for albuminuria were assessed using the TyG index tertiles, the highest tertile with RHF was significantly associated with an increased risk of albuminuria compared to the lowest tertile (AOR in Tertile 3, 5.25; 95% CI, 1.82–15.14; *p =* 0.002). However, in participants without RHF, no significant associations were observed between the TyG index and the risk of albuminuria. We further constructed six groups by combining tertiles of TyG index and the presence or absence of RHF and then evaluated the risk of albuminuria among the six groups (Figure 3). With the lowest tertile of TyG index without RHF as the reference group, the adjusted odds for albuminuria were progressively increased in the higher tertiles with RHF groups (AOR in Tertile 2 with RHF, 2.40; 95% CI, 1.15–5.00; *p* < 0.001 and AOR in Tertile 3 with RHF, 5.77; 95% CI, 3.04–10.95; *p* < 0.001, Table 5). 

Additionally, to further confirm the potential mediation effect of RHF on the association between high TyG index and the risk of albuminuria, we performed mediation analysis, as presence of RHF showed a significant interaction between TyG index and the risk of albuminuria (*p* for interaction < 0.001). In the adjusted multivariable model, the relation between the TyG index and albuminuria was fully mediated by RHF (Figure 4). In particular, the indirect effect of RHF significantly eliminated the significance of the residual direct effect of TyG index on albuminuria (*p* = 0.12 for direct effect), suggesting that RHF mediated 54.2% of the total effect (95% CI of indirect effect, 0.26–0.79; *p* < 0.001). 

Finally, we evaluated the predictive value of the TyG index for risk of albuminuria using ROC curve analysis in participants with or without RHF. After incorporating the TyG index into the basic model including tradition risk factors for albuminuria, AUC was significantly improved only in participants with RHF (AUC, 0.823; 95% CI, 0.765–0.880 in the basic model and AUC, 0.849; 95% CI, 0.794–0.904 in the TyG index-added model; *p* = 0.03) (Table 6 and Appendix A).

### 3.5. Sensitivity Analyses

First, to validate the impact of TyG index on albuminuria and RHF, we examined the interaction term of potent variables which can cause variation in the risk of albuminuria and RHF. Subgroups stratified according to diabetes (no vs. yes), smoking (no vs. yes), and BMI (<25 vs. BMI ≥ 25 kg/m^2^) showed significant interactions with TyG index for the risk of albuminuria (*p* for interactions < 0.05 for all of three variables, Appendix A). The participants with diabetes, smoking, and high BMI (≥25 kg/m^2^) showed a significant association between TyG index and an increased risk of albuminuria after adjustment for confounding variables (AOR, 2.94; 95% CI, 1.40–6.18; *p* = 0.004 in diabetes; AOR, 1.97; 95% CI, 1.45–2.68; *p* < 0.001 in smoking; AOR, 2.16; 95% CI, 1.52–3.06, *p* < 0.001 in BMI ≥ 25 kg/m^2^). However, in the aspect of the risk of RHF, diabetes, smoking, and BMI showed no significant interactions with TyG index (*p* for interaction > 0.05 for all of three variables), suggesting that TyG index was independently associated with the risk of RHF (Appendix A).

Second, to further investigate whether the TyG index with presence of RHF effects the risk of albuminuria according to the glucose levels (prediabetes or diabetes), we performed logistic regression analysis between the combination of TyG index and RHF and the risk of albuminuria in each group of glucose levels. In multivariable logistic regression models, there was a significant association between high TyG index with RHF and the risk of albuminuria in both prediabetes and diabetes groups (Appendix A). The participants showed higher insulin resistance and increased risk of albuminuria as the TyG index levels increased in those with RHF (AOR in prediabetes, 2.05; 95% CI, 1.16–3.65; *p* = 0.01 and AOR in diabetes, 5.32; 95% CI, 1.34–20.75; *p* = 0.02). In contrast, there were no significant associations between TyG index and the risk of albuminuria in those without RHF in the prediabetes or diabetes groups (AOR in prediabetes, 1.64; 95% CI, 0.72–3.72; *p* = 0.24 and AOR in diabetes, 3.00; 95% CI, 0.63–14.25; *p* = 0.17).

Lastly, when the definition of RHF using a higher threshold of eGFR with residuals >95th percentile after adjusting for age, sex, weight, and height was used and analyzed as a study endpoint, similar results were observed: high TyG index was associated with an increased risk of RHF (AOR per 1.0 increased in TyG index level, 2.08; 95% CI, 1.68–2.58; *p* < 0.001 and AOR, 2.44; 95% CI, 1.66–3.60; *p* < 0.001 in the highest tertile of TyG index). When the association between high TyG index with or without RHF defined by eGFR with residuals >95th percentile after adjusting for covariates and the risk of albuminuria was evaluated, high TyG index with RHF showed an increased risk of albuminuria (AOR per 1.0 increase in TyG index, 2.04; 95% CI, 1.06–3.91; *p* = 0.03 and AOR, 3.55; 95% CI, 1.65–15.11; *p* = 0.04 in the highest tertile of TyG index) (Appendix A).

## 4. Discussion

In this cohort of young adults, high TyG index was associated with the risk of albuminuria and RHF. However, an increased risk of albuminuria associated with a high TyG index was only observed in participants with RHF as opposed to those without RHF. Furthermore, participants with concurrently high TyG index and RHF had a stronger association with the risk of albuminuria. The mediation effect of RHF accounted for 54.2% of the relation between the TyG index and albuminuria. Incorporating the TyG index into the traditional risk factor model with RHF significantly improved the predictive value for the risk of albuminuria. In our sensitivity analysis, participants with diabetes, smoking, and high BMI revealed a significant association with high TyG index and the risk of albuminuria. However, TyG index independently increased the risk of RHF regardless of diabetes, smoking, or BMI. Furthermore, the participants with higher insulin resistance showed an increasing risk of albuminuria with high TyG index in those with RHF, whereas those without RHF did not.

TyG index is a surrogate marker for insulin resistance in the younger population, similarly to adults [25,26,27,28]. High TyG index is associated with increased cardiometabolic risks in children to adolescents [29]. Recent studies showed that high TyG index induces kidney dysfunction by enhancing insulin resistance and related metabolic disorders in adults [20,21]. However, the association between high TyG index and kidney disease is poorly understood in young adults. In the present study, as with the previous reports, participants with high TyG index showed higher BMI, BP, fasting glucose levels, and poor lipid profiles compared with those with low TyG index. Moreover, participants with high TyG index were more likely to have previous histories of the metabolic disorder, including hypertension, diabetes, and dyslipidemia. Notably, participants with high TyG index showed significantly higher risks for RHF and albuminuria. In particular, the risk of albuminuria was stronger in participants with concurrent high TyG index and RHF. The mechanism linking high TyG index, RHF, and albuminuria remains unclear. However, our findings suggest that high TyG index and associated metabolic disturbances may exacerbate insulin resistance, which in turn may predispose one to kidney disease.

The increasing incidence of obesity, hypertension, and diabetes among young adults is a growing public health concern worldwide [5]. This phenomenon eventually leads to insulin resistance, which increases the risk of adverse kidney outcomes [30]. This is supported by two observational studies in adolescents with T2D [17,18]. In both RESISTANT and TODAY studies, young adults with T2D showed that insulin resistance was strongly associated with RHF. Notably, insulin resistance was the most significant determinant for RHF rather than traditional risk factors, including glycemic or BP control status or dyslipidemia. Insulin resistance plays a pathogenic role in kidney dysfunction through several mechanisms, not only among adults with diabetes but also among non-diabetics [31,32]. The potential mechanisms are the result of complex effects of insulin resistance and related metabolic disorders, including obesity, elevated BP or blood glucose levels, and dyslipidemic status [33]. These metabolically abnormal features result in renal hemodynamic changes, an increase in intra-glomerular pressure, and glomerular hypertrophy, which all promote RHF. Of note, RHF is a well-known predisposing risk factor to irreversible kidney damage contributing to initiation and progression to CKD in diabetes. Recent studies suggest that RHF serves as an early risk factor for kidney damage even in non-diabetics, although the mechanism by which RHF induces progressive CKD remains unclear [34,35].

Nevertheless, growing evidence suggests that insulin resistance or its related metabolic disturbance is associated with RHF and its attribution to the development of CKD in non-diabetic adults and the adolescent population [36,37]. In a prospective cohort study with 1261 middle-aged (50 to 62 years old) participants without overt diabetes, prediabetes was independently associated with increased odds for RHF and development of albuminuria [38]. The results from KNHANES data with 15,918 adult participants aged more than 18 years showed that each component of metabolic syndrome was associated with a higher risk of RHF [36]. Furthermore, RHF was independently associated with albuminuria. In a cohort study using the National Health and Nutrition Examination Survey data, 8793 US adolescents aged 12 to 17 without diabetes showed that insulin resistance (when using the homeostatic model assessment of insulin resistance) was associated with RHF [39]. Hypertriglyceridemia and hyperinsulinemia were also significantly associated with RHF in this study. The prevalence of insulin resistance is increasing even in young adults without overt diabetes due to rising obesity rates resulting from lifestyle changes [37,40]. Given the progressive nature of CKD, the number of young adults with CKD or albuminuria caused by increased insulin resistance is likely to increase. Based on these findings, early detection of an increase in insulin resistance in young adults may prevent RHF and progressive overt nephropathy in the future. Notably, the results of this study suggest that high TyG index with RHF can be used as an early marker for identifying worsening kidney function.

This study has several limitations. First, due to the cross-sectional observational study design, a causal relationship and mediation effect between the TyG index, RHF, and their effect on albuminuria development cannot be fully determined. Although a cross-sectional mediation analysis was performed to clarify the relationship between the TyG index, RHF, and the risk of albuminuria, a further longitudinal study is warranted to confirm causality. Second, due to the limitation of data, other insulin resistance indicators such as HOMA-IR or serum insulin levels were not measured and compared to the TyG index. However, previous studies have indicated that the TyG index is a reliable marker for insulin resistance compared to other indicators, particularly in young adults [27,28,41,42]. Third, GFR was not directly measured, and the definition of RHF was estimated using the cutoff value of eGFR. More specifically, RHF was defined by using the adjusted eGFR based on linear regression methods, as described in a previous study [11]. Furthermore, we used RHF defined by a higher threshold of eGFR with residuals >95th percentile after adjusting for age, sex, weight, and height, and the main results were consistently observed. A clear definition of RHF remains controversial in children and young adults, and further studies to validate the definition of RHF using direct assessment of GFR are warranted. Finally, the association between high TyG index, RHF, and advanced CKD could not be determined in this study because the participants consisted of relatively healthy young adults, and most of them had albuminuria, representing early kidney injury. It is necessary to evaluate how kidney function changes in individuals with a high TyG index and RHF at a younger age compared to the normal population in future research, with long-term follow ups.

## 5. Conclusions

High TyG index was associated with an increased risk of RHF and albuminuria in young adults. High TyG index with the presence of RHF was the strongest risk factor for albuminuria in this study. TyG index, a surrogate marker of insulin resistance, may serve as an early indicator of kidney damage. These findings suggest that early identification of high TyG index with RHF may help prevent future development of CKD in relatively healthy and young individuals.

## Figures and Tables

**Figure 1 jcm-11-06419-f001:**
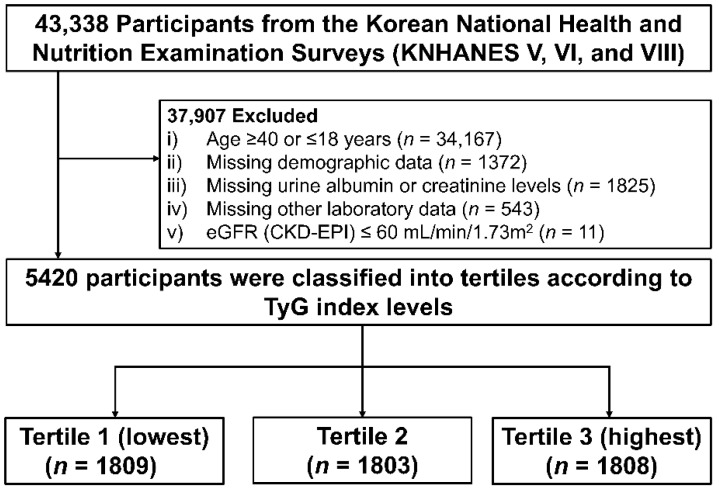
Flow chart of study population. Abbreviations: KNHANES—Korea National Health and Nutrition Examination Survey; TyG index—triglyceride-glucose index.

**Figure 2 jcm-11-06419-f002:**
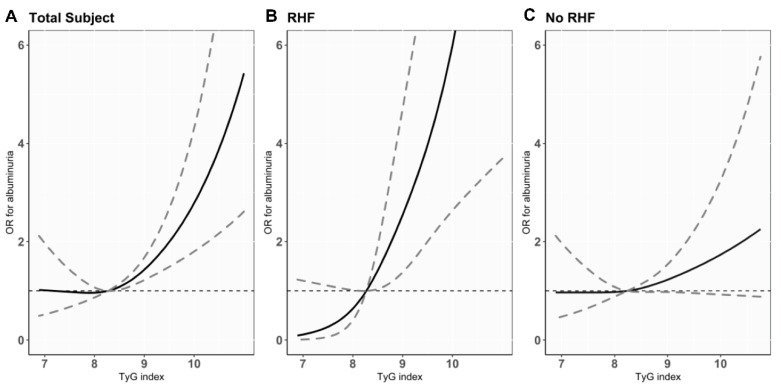
Cubic spline curves for the risk of albuminuria according to TyG index in total study participants (**A**) and participants with RHF (**B**) or without RHF (**C**). Black lines show ORs per 1.0 increase of TyG index and gray dashed lines represent 95% CIs. Abbreviations: TyG index—triglyceride glucose index; RHF—renal hyperfiltration; OR—odds ratio; CI—confidence interval.

**Figure 3 jcm-11-06419-f003:**
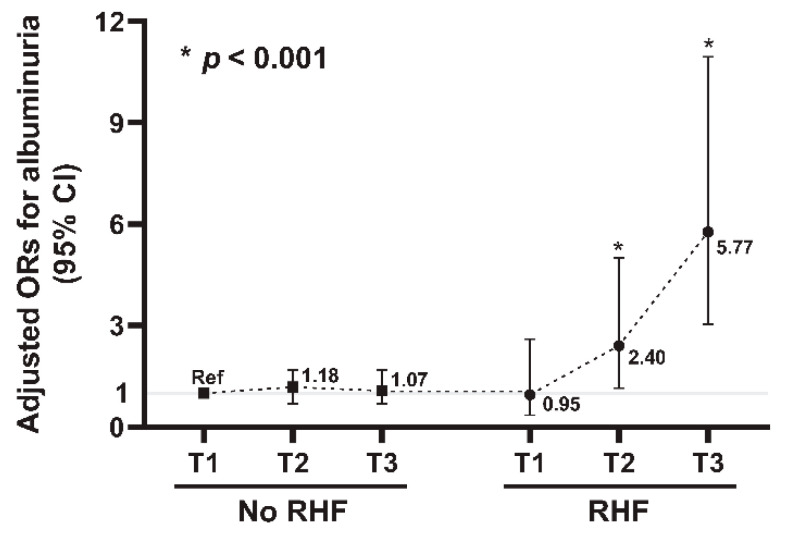
Association between the six groups grouped according to TyG index tertile with or without RHF and albuminuria. Each number indicates ORs for albuminuria and the error bars represent 95% CIs. Abbreviations: TyG index—triglyceride glucose index; RHF—renal hyperfiltration; OR—odds ratio; CI—confidence interval.

**Figure 4 jcm-11-06419-f004:**
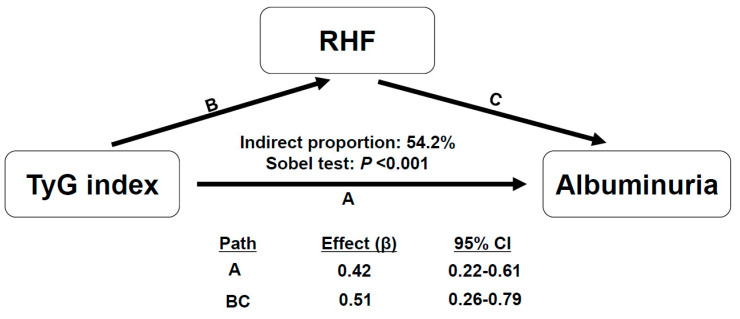
Mediation analysis of the association between TyG index and albuminuria mediated by RHF. Path effect is reported as the estimated regression coefficients (β). Residual direct effect is labeled as path A and indirect effects are labeled with relevant groups of path BC. Analyses are adjusted for age; sex; smoking and alcohol status; education and income levels; BMI; SBP; hemoglobin; eGFR (CKD-EPI); and past history of HTN, DM, and dyslipidemia Abbreviations: TyG index—triglyceride glucose index; RHF—renal hyperfiltration; CI—confidence interval; BMI—body mass index; SBP—systolic blood pressure; eGFR—estimated glomerular filtration rate; CKD-EPI—CKD Epidemiology Collaboration; HTN—hypertension; DM—diabetes mellitus.

**Table 1 jcm-11-06419-t001:** Baseline characteristics.

	TyG Index Tertile			
Characteristics	Tertile 1 (Lowest) (*n* = 1809)	Tertile 2 (*n* = 1803)	Tertile 3 (Highest) (*n* = 1808)	*p*
**Demographic data**				
Age, years, mean (SD)	29.4 ± 6.2	30.3 ± 6.1	32.2 ± 5.4	<0.001
Male, *n* (%)	462 (25.5%)	820 (45.5%)	1234 (68.3%)	<0.001
Smoking status, *n* (%)	446 (24.7%)	739 (41.0%)	1060 (58.6%)	<0.001
Alcohol status, *n* (%)	804 (44.4%)	929 (51.5%)	1133 (62.7%)	<0.001
Education, *n* (%)				<0.001
Low	344 (19.0%)	451 (25.0%)	533 (29.5%)	
High	1465 (81.0%)	1352 (75.0%)	1275 (70.5%)	
Income, *n* (%)				0.02
Low	837 (46.3%)	889 (49.3%)	921 (50.9%)	
High	972 (53.7%)	914 (50.7%)	887 (49.1%)	
BMI, kg/m^2^, mean (SD)	21.4 ± 2.9	22.6 ± 3.4	25.5 ± 4.0	<0.001
SBP, mmHg, mean (SD)	105.4 ± 10.0	109.1 ± 11.3	115.1 ± 13.0	<0.001
DBP, mmHg, mean (SD)	69.5 ± 8.3	72.2 ± 9.2	77.7 ± 11.0	<0.001
**Comorbidities**				
Hypertension, *n* (%)	32 (1.8%)	85 (4.7%)	265 (14.7%)	<0.001
Diabetes, *n* (%)	3 (0.2%)	7 (0.4%)	109 (6.0%)	<0.001
Dyslipidemia, *n* (%)	6 (0.3%)	16 (0.9%)	68 (3.8%)	<0.001
**Laboratory data**				
TyG index, mean (SD)	7.7 ± 0.3	8.3 ± 0.2	9.1 ± 0.5	<0.001
eGFR, mL/min/1.73 m^2^, mean (SD)	110.6 ± 11.9	108.4 ± 12.7	106.1 ± 13.2	<0.001
Microalbuminuria, *n* (%)	44 (2.4%)	64 (3.5%)	125 (6.9%)	<0.001
Hemoglobin, g/dL, mean (SD)	13.5 ± 1.5	14.2 ±1.6	14.9 ± 1.6	<0.001
HbA1c, %, mean (SD)	5.3 ± 0.3	5.4 ± 0.4	5.7 ± 0.8	<0.001
Fasting plasma glucose, g/dL, mean (SD)	86.3 ± 6.9	89.8 ± 7.7	97.6 ± 24.5	<0.001
Total cholesterol, mg/dL, mean (SD)	166.9 ± 27.1	177.9 ± 29.1	197.3 ± 36.5	<0.001
LDL-C, mg/dL, mean (SD)	97.3 ± 23.7	106.7 ± 27.3	116.5 ± 33.2	<0.001
HDL-C, mg/dL, mean (SD)	59.4 ± 11.1	53.8 ± 11.4	46.5 ± 11.2	<0.001
Triglyceride, mg/dL, mean (SD)	52.0 ± 11.8	89.1 ± 14.8	206.4 ± 140.6	<0.001

Data are presented as mean (SD), or number (%). Tertile 1: 5.89 ≤ TyG index ≤ 8.02; Tertile 2: 8.02 ≤ TyG index ≤ 8.55; Tertile 3: 8.55 ≤ TyG index ≤ 12.07; microalbuminuria, UACR ≥ 30 mg/g Cr. Abbreviations: BMI—body mass index; SBP—systolic blood pressure; DBP—diastolic blood pressure; eGFR—estimated glomerular filtration rate; LDL-C—low-density lipoprotein cholesterol; HDL-C—high-density lipoprotein cholesterol.

**Table 2 jcm-11-06419-t002:** Logistic regression analyses for albuminuria according to TyG index.

	Prevalence * *n* (%)	Model 1		Model 2		Model 3	
Variable		OR (95% CI)	*p*	OR (95% CI)	*p*	OR (95% CI)	*p*
**TyG index** (per 1.0 increase)	233 (4.3)	2.41 (2.03–2.86)	<0.001	2.63 (2.18–3.18)	<0.001	1.56 (1.24–1.95)	<0.001
**Tertile of TyG index**							
Tertile 1 (lowest)	44 (2.4)	(Reference)					
Tertile 2	64 (3.5)	1.64 (1.12–2.40)	0.01	1.66 (1.13–2.44)	0.01	1.49 (1.01–2.21)	0.04
Tertile 3 (highest)	125 (6.9)	3.05 (2.15–4.33)	<0.001	3.08 (2.12–4.49)	<0.001	1.65 (1.08–2.52)	0.02

* *p* < 0.001. Model 1: Unadjusted model. Model 2: Adjusted for age and sex. Model 3: Adjusted for age; sex; smoking and alcohol status; education and income levels; BMI; SBP; hemoglobin; eGFR(CKD-EPI); and past history of HTN, DM, and dyslipidemia. Abbreviations: TyG—triglyceride-glucose; OR—odds ratio; CI—confidence interval; BMI—body mass index; SBP—systolic blood pressure; eGFR—estimated glomerular filtration rate; CKD-EPI—CKD Epidemiology Collaboration; HTN—hypertension; DM—diabetes mellitus.

**Table 3 jcm-11-06419-t003:** Logistic regression analyses for RHF according to TyG index.

	Prevalence * *n* (%)	Model 1		Model 2		Model 3	
Variable		OR (95% CI)	*p*	OR (95% CI)	*p*	OR (95% CI)	*p*
**TyG index** (per 1.0 increase)	543 (10)	2.03 (1.79–2.30)	<0.001	1.49 (1.30–1.72)	<0.001	1.56 (1.32–1.84)	<0.001
**Tertile of TyG index**							
Tertile 1 (lowest)	107 (5.9)	(Reference)					
Tertile 2	161 (8.9)	1.56 (1.21–2.01)	<0.001	1.14 (0.88–1.48)	0.33	1.19 (0.91–1.55)	0.21
Tertile 3 (highest)	275 (15.2)	2.84 (2.24–3.58)	<0.001	1.56 (1.21–2.02)	<0.001	1.73 (1.31–2.30)	<0.001

* *p* < 0.001, Model 1: Unadjusted model. Model 2: Adjusted for age and sex. Model 3: Adjusted for age; sex; smoking and alcohol status; education and income levels; BMI; SBP; hemoglobin; albuminuria; and past history of HTN, DM, and dyslipidemia. Abbreviations: RHF—renal hyperfiltration; TyG—triglyceride-glucose; OR—odds ratio; CI—confidence interval; BMI—body mass index; SBP—systolic blood pressure; HTN—hypertension; DM—diabetes mellitus.

**Table 4 jcm-11-06419-t004:** Association between albuminuria and TyG index with or without RHF.

	Prevalence * *n* (%)	Model 1		Model 2		Model 3	
Variable		OR (95% CI)	*p*	OR (95% CI)	*p*	OR (95% CI)	*p*
**With RHF**							
**TyG index** (per 1.0 increase)	64 (11.8)	4.22 (2.90–6.15)	<0.001	4.20 (2.86–6.16)	<0.001	2.75 (1.74–4.35)	<0.001
**Tertile of TyG index**							
Tertile 1 (lowest, *n* = 181)	5 (2.8)	(reference)					
Tertile 2 (*n* = 180)	14 (7.8)	2.97 (1.05–8.42)	0.04	2.89 (1.02–8.23)	0.04	2.44 (0.82–7.30)	0.11
Tertile 3 (highest, *n* = 182)	45 (24.7)	11.56 (4.47–29.9)	<0.001	10.90 (4.16–28.58)	<0.001	5.25 (1.82–15.14)	0.002
**Without RHF**							
**TyG index** (per 1.0 increase)	169 (3.5)	1.65 (1.33–2.05)	<0.001	1.93 (1.52–2.45)	<0.001	1.23 (0.92–1.65)	0.16
**Tertile of TyG index**							
Tertile 1 (lowest, *n* = 1627)	43 (2.6)	(reference)					
Tertile 2 (*n* = 1625)	53 (3.3)	1.24 (0.83–1.87)	0.09	1.36 (0.90–2.05)	0.15	1.25 (0.82–1.90)	0.31
Tertile 3 (highest, *n* = 1625)	73 (4.5)	1.73 (1.18–2.54)	0.005	2.10 (1.39–3.18)	<0.001	1.18 (0.74–1.91)	0.49

* *p* < 0.001. Model 1: Unadjusted model. Model 2: Adjusted for age and sex. Model 3: Adjusted for age; sex; smoking and alcohol status; education and income levels; BMI; SBP; hemoglobin; eGFR(CKD-EPI); and past history of HTN, DM, and dyslipidemia. Abbreviations: CKD—chronic kidney disease; RHF—renal hyperfiltration; TyG—triglyceride-glucose; OR—odds ratio; CI—confidence interval; BMI—body mass index; SBP—systolic blood pressure; eGFR—estimated glomerular filtration rate; CKD-EPI—CKD Epidemiology Collaboration; HTN—hypertension; DM—diabetes mellitus.

**Table 5 jcm-11-06419-t005:** Subgroup analysis stratified by the combination of RHF and TyG index.

	Prevalence of Albuminuria *, *n* (%)	Model 1		Model 2		Model 3	
Variable		OR (95% CI)	*p*	OR (95% CI)	*p*	OR (95% CI)	*p*
**RHF + Tertile of TyG**							
No RHF + T1 (*n* = 1627)	43 (2.6)	(Reference)					
No RHF + T2 (*n* = 1625)	53 (3.3)	1.24 (0.83–1.87)	0.30	1.33 (0.88–2.01)	0.18	1.18 (0.78–1.80)	0.43
No RHF + T3 (*n* = 1625)	73 (4.5)	1.73 (1.18–2.54)	0.005	1.99 (1.32–2.99)	0.001	1.07 (0.68–1.68)	0.78
RHF + T1 (*n* = 181)	5 (2.8)	1.05 (0.41–2.68)	0.92	1.33 (0.51–3.45)	0.56	0.95 (0.35–2.60)	0.92
RHF + T2 (*n* = 180)	14 (7.8)	3.11 (1.67–5.80)	<0.001	3.87 (2.03–7.40)	<0.001	2.40 (1.15–5.00)	<0.001
RHF + T3 (*n* = 182)	45 (24.7)	12.10 (7.69–19.03)	<0.001	15.47 (9.36–25.58)	<0.001	5.77 (3.04–10.95)	<0.001

* *p* < 0.001, Model 1: Unadjusted model, Model 2: Adjusted for age and sex, Model 3: Adjusted for age; sex; smoking and alcohol status; education and income levels; BMI; SBP; hemoglobin; eGFR(CKD-EPI); and past history of HTN, DM, and dyslipidemia Abbreviations: RHF—renal hyperfiltration; TyG index—triglyceride-glucose index; OR—odds ratio; CI—confidence interval; BMI—body mass index; SBP—systolic blood pressure; eGFR—estimated glomerular filtration rate; CKD-EPI—CKD Epidemiology Collaboration; HTN—hypertension; DM—diabetes mellitus.

**Table 6 jcm-11-06419-t006:** Comparison of predictive power for the risk of albuminuria with TyG index with or without RHF.

Variable	AUC	95% CI	*p*-value
**With RHF**			
Basic model *	0.823	0.765–0.880	
Basic model + TyG index	0.849	0.794–0.904	0.03 ^#^
**Without RHF**			
Basic model *	0.673	0.622–0.717	
Basic model + TyG index	0.673	0.629–0.717	0.94 ^#^

Comparison was performed using ROC analysis. * Basic model includes traditional risk factors for CKD such as age; sex; smoking and alcohol status; education and income levels; BMI; SBP; hemoglobin; eGFR; and past history of HTN, DM, and dyslipidemia. ^#^
*p* value for the improvement of predictive power of basic model + TyG index compared with the basic model. Abbreviations: CKD—chronic kidney disease; RHF—renal hyperfiltration; TyG index—triglyceride-glucose index; ROC—receiver operating characteristic; AUC—area under curve; CI—confidence interval; BMI—body mass index; SBP—systolic blood pressure; eGFR—estimated glomerular filtration rate; HTN—hypertension; DM—diabetes mellitus.

## Data Availability

This data can be found here: [https://knhanes.kdca.go.kr/knhanes/].

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
