# Peer review of "High Triglyceride-Glucose Index with Renal Hyperfiltration and Albuminuria in Young Adults: The Korea National Health and Nutrition Examination Survey (KNHANES V, VI, and VIII)"

_jcm, 2022, doi:10.3390/jcm11216419_

Round 1

Reviewer 1 Report

The manuscript entitled “High Triglyceride-Glucose Index with Renal Hyperfiltration and Albuminuria in Young Adults: the Korea National Health and Nutrition Examination Survey (KNHANES V, VI, and VIII)" by Oh et al. (jcm-1972327) evaluates the association of insulin resistance, as assessed by TyG, with renal hyperfiltration and albuminuria in young adults. The paper is well written, the analysis is well done, and results clearly presented.

My main concern is why the ‘last’ results are placed in the supplementary data. FigS1 and Tables S1 and S2 should be placed in the main body of the manuscript as they are relevant for the conclusions.

Furthermore, the conclusion in the Abstract should be revised and go more in accordance to the conclusions in the discussion.

Reviewer 2 Report

Oh DH et al. explored the relationship between the TyG index, a marker of insulin resistance, renal hyperfiltration (RHF) and albuminuria among young adults participating in KNHANES between 2011 and 2019. They found that a high TyG index is associated with higher odds of albuminuria or RHF, and that participants with both high TyG and RHF had the highest risk for albuminuria.

I offer the following suggestions for consideration:

1.      Under paragraph 3.2 Association of TyG index with albuminuria, It states that the proportion of persons with albuminuria was higher in the highest TyG tertile, however, according to table 2 there was no difference in mean ACR between TyG tertiles. Please reconcile for clarity.

2.      Please indicate the ranges for the TyG tertiles, either in table 1 or text.

3.       As a sensitivity analysis, the authors may consider also defining RHF as the value 2 SDs above the mean eGFR for this population or the 95th percentiles, which are more commonly used definitions and give a better understanding of the absolute values as predictors of outcome.

4.      In table 2, model 3, by including TyG in the same model with diabetes, BMI, and smoking, TyG becomes an intermediate variable, i.e., one that causes variation in the dependent variable (UACR) and is itself caused to vary by the independent variables listed above. Therefore, a mediation effect should be also explored for TyG. Alternatively, the analysis could be restricted to more parsimonious models. Same holds for the other fully adjusted models.

Round 2

Reviewer 2 Report

The authors' revisions are satisfactory.